# Asymptomatic carriage of *Plasmodium falciparum* in children no longer targeted for seasonal malaria chemoprevention and with a history of exposure to this strategy: A cross sectional study in southern Senegal

Isaac Akhénaton Manga[1]*, Abdoulkarim Mhadji[1◉], Aminata Lam[1‡], Marie Pierre Diouf[2], Pab Carole Minlekib[1], Fassiatou Tairou[1], Amadou Seck[2], Babacar Faye[1], Magatte Ndiaye[1‡], Jean Louis Abdourahim Ndiaye[1,2◉]

1 Department of Parasitology-Mycology/Faculty of Medicine, Pharmacy and Odontology, Cheikh Anta Diop University, Dakar, Senegal, 2 Department of Parasitology-Mycology/Medical Biology, UFR Santé/Iba Der Thiam University, Thiès, Senegal

◉ These authors contributed equally to this work.
‡ AL and MN also contributed equally to this work.
* akhenmanga@yahoo.fr, isaacakhenaton1.manga@ucad.edu.sn

## Abstract

Seasonal malaria chemoprevention (SMC) has been implemented in southern part of Senegal since 2013, targeting children 3–120 months old. This strategy is often evaluated among the population targeted for the intervention. This study was conducted to determine the prevalence of malaria in children who recently aged out of the target age range for SMC in Senegal. The study was conducted between September and December 2016 in Kédougou, Kolda and Sédhiou, located in southern Senegal where SMC is implemented. A questionnaire was administered to each participant to collect information on his history of SMC exposure and a blood sample collected on Whatman filter paper. The *Pf18S* gene was amplified by Real Time-PCR to detect *Plasmodium falciparum*. A total of 226 children between 11 and 14 years old were included in the study. The mean age was 11.9 (±0.8) years with a sex ratio (M/F) of 1.05. The overall malaria prevalence was 19.47%. *Plasmodium falciparum* parasite carriage was higher in children who had recently aged out of SMC eligibility (6.6% in 11-year-olds and 9.3% in 12-year-olds). In 2013, the prevalence was higher among children who had completed two cycles (5.3%). However, it was higher among those who had taken three cycles in 2014 (13.7%) and 2015 (5.8%). This study showed that the prevalence of *P. falciparum* was higher in children who had taken SMC in 2013 and 2014, with a prevalence of 6.6%. Children exposed in 2013, 2014 and 2015 had a prevalence of 2.65%. No parasite carriage was observed in children exposed in 2013, 2014, 2015 and 2016. Children protected by SMC become potential malaria parasites reservoirs when they leave the target age range. There is an urgent need of strategies, other than bed nets to continue protecting these populations.

**Data availability statement:** All relevant data are within the paper and its Supporting Information files.

**Funding:** The author(s) received no specific funding for this work.

**Competing interests:** The authors have declared that no competing interests exist.

## Introduction

Malaria remains an endemic public health problem in Senegal. In 2021, the country recorded 536,850 cases and 399 deaths due to malaria [1]. Malaria distribution is heterogeneous in the country, with the southeast regions of Kolda, Kedougou and Tambacounda bearing the highest burden. In 2021, 78.5% of confirmed malaria total cases and 87.9% of cases in children under five in 2021 occurred in these regions [1]. Despite this high malaria prevalence in the southeast, a drastic reduction in malaria prevalence is noted country-wide since the early 2000s. From 2015 to 2020, a significative reduction (21.6%) of malaria morbidity was observed from 4.86% to 3.81% respectively. The same trend was noted from 2015 (3.52%) to 2020 (2,07%) with a reduction of 41.2%. Among children under 5, there has been a significant reduction of 23% (2.27% in 2015 to 1.75% in 2020) and 53.6% (3.80% in 2015 to 1.76% in 2020) in malaria morbidity and proportional mortality respectively [1]. This reduction in malaria morbidity and mortality has been attributed to the implementation of principal strategies recommended by the World Health Organization (WHO). These strategies include seasonal malaria chemoprevention (SMC), which consists of administration of a monthly full dose of Sulfadoxine-pyrimethamine plus amodiaquine (SP-AQ) to prevent malaria during the high period of malaria transmission. Since March 2012, SMC has been recommended by WHO for children aged 3-59 months living in malaria-endemic areas with high to moderate seasonal malaria transmission [2]. Several operational research studies on SMC have shown the strategy to be cost-effective [3,4] and well-tolerated [5,6]. Since 2013, SMC with SP-AQ is implemented in Senegal [7] among children aged 3–120 months. This choice was motivated by the shift of malaria burden from under 5 to 5–10 years age group [8–10]. The use of community health workers and the door-to-door strategy for SMC administration has facilitated the high coverage of the intervention in the target group. In 2020, 85.8% of the target population received a full course of three doses of SP-AQ [1]. Evaluation of the impact of this strategy, recommended by WHO, has shown that SMC still be effective [11] and reduces the prevalence of malaria in the target population [12]. However, there is limited information on malaria prevalence on children over 10 years old particularly after their exit from the SMC target. In this context, this study was conducted to assess *Plasmodium falciparum* malaria carriage in children with a history of exposure to SMC.

## Methodology

### Type, period, population and study site

The study was conducted in the regions where the SMC is implemented in south-eastern Senegal (Kédougou, Kolda, Sédhiou and Tambacounda). In these regions, malaria is one of the main reasons for consultations. The proportional morbidity due to malaria in 2021 was 17,668 cases in Sédhiou, i.e., 3.3% of the national total, 105,694 cases in Kédougou, i.e., 19.6%, and 181,999 cases in Kolda, i.e., 33.9% [1]. They also met all WHO criteria for eligibility for SMC [2].

The study was part of a project to evaluate the impact of SMC administered as a mass campaign in southern Senegal. This project had several components, including an evaluation of the impact of SMC on immunity. To do this, a case-control study was conducted among children who were no longer targeted by SMC. A case was defined as any child aged 11-14 years seen at a health post in our study area who tested positive for malaria. A control was defined as a child of the same age and sex living in the same or a neighboring compound as the case who also had to have a negative malaria RDT and show no clinical signs (i.e., be in apparent good health). This case-control study took place between 7 September and 17 December 2016. To detect about 40% of malaria cases among older children in our study area, our sample size was estimated at about 500 cases and controls using Epi info 7.1.3.3 software (RRID:SCR_021682).

To answer the research question of our study, we were only interested in the controls recruited to assess the impact of SMC on immunity in the project. All controls with at least one history of exposure to SMC were included in our study. Free and informed consent and assent from the child's parent or legal guardian were also criteria for inclusion. Therefore, any control who had never been exposed to SMC and/or had a positive RDT for *P. falciparum* was not included.

### Field activities

A questionnaire was administrated to collect socio-demographic data (age, sex, village, health district, etc.) and the history of exposure to SMC from 2013 to 2016. Blood samples were blotted into Whatman filter paper and carefully packed in zip log bag with silica to avoid desiccation. All biological samples were shipped in the Parasitology-Mycology laboratory at the Faculty of Medicine UCAD for the molecular diagnosis of malaria by real-time PCR.

### Laboratory activities

*Plasmodium* deoxyribonucleic acid (DNA) was extracted from dried blood spots using the Chelex-100 method described by Wooden [13]. Extracted DNA was used to amplify the *P.f18S* gene of *Plasmodium falciparum* by real-time PCR (RT-PCR). *P. falciparum* DNA amplification was performed using the pan-primer Plasmo 2 REV (5'-AAC CCA AAG ACT TTG ATT TCT CAT AA-3'); Fal-FORW (5'-CCG ACT AGG TGT TGG ATG AAA GTG TTA A-3') and the Falcprobe probe (5'Quasar 670-AGCAATCTAAAAGTCACCTCGAAAGATGACT-BHQ-2 3') [14]. Each of these primers or probe was first diluted to the 10th. A reaction mixture consisted of 5 μl master mix (TaqMan™ Fast Advanced Master Mix); 0.2 μl sense primer; 0.2 μl anti-sense primer; 0.1 μl probe and 3.5 μl sterile water, was made for the amplification. In each well of a 96-well polymerase chain reaction (PCR) plate, 9 μl of the reaction mixture and 1 μl of extracted DNA was added. 3D7 strain was used as positive control; sterile water and reaction mix were used as negative controls. The microplate was sealed with adhesive film (Thermo scientific AB-0558, ref: 124730) and centrifuged before being placed in the BIO-RAD CFX-96 Real Time PCR thermocycler for DNA amplification according to the following program [S1 Table].

### Interpretation of RT-PCR results

A sample was considered positive when it presented an amplification curve identical to the positive control and his cycle of quantification (Cq) curve appear less than 35 cycles. Samples with Cq greater than 35 was consider to be negative.

### Data entry and analysis

Data obtained from the questionnaires and Bio Rad CFX Manager 3.1 RT-PCR software were entered into Microsoft Office Excel 2013 and analyzed using R software (R version 4.1.0). Qualitative variables (sex, age group, history of exposure to SMC, parasite carriage) were described as numbers and percentages with 95% confidence intervals (95% CI). Age of the participants, which was the only quantitative variable, was described using mean with his standard deviation.

### Ethical considerations

The study was approved by the Senegalese National Health Research Ethics Committee under approval number CNERS SEN13/57. Free and informed written consent was obtained from parents or legal representatives. The assent of all the children included had also been obtained

before the start of the study. To ensure confidentiality, an identification code was assigned to each child recruited.

## Results

### Sociodemographic characteristics of the study population and history of exposure to SMC

A total of 226 participants aged 11-14 years were recruited [S1 Fig]. More than half (63.7%; 95% CI = 59.7-67.7) were recruited in Kolda. The *sex ratio* (M/F) was 1.05 and gender information was missing for 2.2% of them. The mean age of the children was 11.9 years (±0.8) and most of them were aged 11 and 12, with respectively 35.4% (95% CI = 29.2-41.6) and 46% (95% CI = 39.5-52.5). Age information was missing for 0.4% (95% CI = 0-1.3) of the children.

Among children included, 33.6% (95% CI = 30.5-36.8) had received SMC in 2013, 87.6% (95% CI = 85.4-89.8) in 2014, and 38.5% (95% CI = 35.3-41.7) in 2015. Children who had received SMC in both 2013 and 2014 represented 23% (95% CI = 17.5-28.5), those who had been exposed in 2013, 2014, and 2015 (3 years) represented 8.4% (95% CI = 4.8-12). Only 0.4% (95% CI = 0-1.3) had received SMC for four consecutive years (2013-2016) [S2 Table].

### Asymptomatic carriage of Plasmodium falciparum

Of the 226 children included in this study, 19.5% (44/226) were *P. falciparum* positive by RT-PCR. Almost all asymptomatic carriers were found in the Kédougou and Kolda regions, with proportions of 50% (95% CI = 42.5-57.5) and 47.7% (95% CI = 36.8-58.6), respectively. Carriage was higher in girls (54.5%; 95% CI = 39.8-69.2) than in boys (43.2%; 95% CI = 28.6-57.8). The majority of positives were 12 years old (47.7%; 95% CI = 32.9-62.5) followed by 11-year-olds (34.1%; 95% CI = 20.1-48.1). Among the positives, 47.7% (95% CI = 32.9-62.5) had taken SMC in 2013; 86.4% (95% CI = 81.2-91.6) in 2014 and 36.4 (95% CI = 29.2-43.6) in 2015. No carriage was noted among children who had taken it in 2016.

Among carriers of *P. falciparum*, 34.1% (95% CI = 27-41.2) had taken SMC both in 2013 and 2014, 13.6% (6/44) (95% CI = 8.4-18.8) in 2013, 2014 and 2015. No parasite carriage was noted among those exposed in 2013, 2014, 2015 and 2016. These differences in prevalence were statistically significant (p = 0.02) [S3Table].

## Discussion

The aim of this study was to assess the asymptomatic carriage of *P. falciparum* malaria parasites in children with a history of SMC who were no longer in its target group. Although some data, such as age and sex, were not available for some of the children included, it was found that 19.5% of them were asymptomatic carriers of *P. falciparum.* Older children and adolescents are therefore an important reservoir of malaria parasites. This was also reported in a study in Senegal, where 24.7% of adolescents were found to be carrying malaria parasites [15]. *P. falciparum* infection in this study was much more common in children who had recently stopped taking SMC, i.e., those aged 11- and 12-years vs 13 and 14, and numerous studies have reported that exposure to chemopreventive malaria drugs could lead to a decline in anti-malarial immunity. Indeed, previous studies conducted in Senegal showed a decrease in the production of anti-Plasmodium antibodies in children who had received chemoprevention [16–18]. The same observation was made in The Gambia in children taking pyrimethamine and dapsone-based chemoprevention [19]. Similar results on the effect of chemoprevention on children's antimalarial immunity were also obtained in Ghanaian children 6 months after SP preventive treatment [20]. In addition to these studies

showing a reduction in anti-Plasmodium antibodies associated with chemoprevention, others have shown that the production of these antibodies increases with age and *P. falciparum* burden [21–23].

At the end of this study, and to better assess the impact of SMC on children who are no longer in the target group, it would be important to compare the kinetics of anti-Plasmodium antibodies between children who are still in the SMC target age group and those who have aged-out. The impact of SMC on this immunity could also be assessed by comparing antibody levels in SMC and non-SMC areas.

## Conclusion

This study found significant asymptomatic *Plasmodium falciparum* carriage in adolescents with a history of exposure to SMC. It was more pronounced in children who had aged out of the target group of this strategy in Senegal. Older children are therefore a major reservoir of parasites, maintaining malaria transmission in the country.

## Supporting information

**S1 Table. Table presenting the Amplification program for the *P. f18S* gene specific for *Plasmodium falciparum*.**
(DOCX)

**S1 Fig. Figure presenting the Flowchart of recruitment and enrolment of children in the study.**
(DOCX)

**S2 Table. Table presenting the distribution of enrolled children according to their socio-demographic characteristics and history of exposure to SMC between 2013 and 2016.**
(DOCX)

**S3 Table. Table presenting the distribution of positive cases by sociodemographic characteristics and SMC exposure history.**
(DOCX)

## Acknowledgments

The authors would like to thank all health workers and the population of the health districts of Kédougou, Vélingara, Kolda and Sedhiou in Senegal. They also thank the staff of the Parasitology-Mycology Department of the Faculty of Medicine, Pharmacy and Odontology of the Cheikh Anta Diop University in Dakar.

Acknowledgments also to Leah F. Moriarty, PhD, MPH, Epidemiologist at Malaria Branch, Centers for Disease Control & Prevention, President's Malaria Initiative (lmoriarty@cdc.gov) for English improvement.

## Author contributions

**Conceptualization:** Isaac Akhenaton Manga, Marie Pierre Diouf, Babacar Faye, Magatte Ndiaye, Jean Louis Abdourahim Ndiaye.

**Data curation:** Isaac Akhenaton Manga, Abdoulkarim Mhadji, Marie Pierre Diouf, Pab Carole Minlekib, Fassiatou Tairou, Amadou Seck, Magatte Ndiaye.

**Formal analysis:** Aminata Lam, Pab Carole Minlekib.

**Investigation:** Fassiatou Tairou, Amadou Seck, Jean Louis Abdourahim Ndiaye.

**Methodology:** Isaac Akhenaton Manga, Abdoulkarim Mhadji, Marie Pierre Diouf, Pab Carole Minlekib, Babacar Faye, Magatte Ndiaye, Jean Louis Abdourahim Ndiaye.

**Project administration:** Jean Louis Abdourahim Ndiaye.

**Resources:** Jean Louis Abdourahim Ndiaye.

**Software:** Isaac Akhenaton Manga, Abdoulkarim Mhadji, Aminata Lam, Amadou Seck, Magatte Ndiaye.

**Supervision:** Isaac Akhenaton Manga, Aminata Lam, Marie Pierre Diouf, Pab Carole Minlekib, Fassiatou Tairou, Amadou Seck, Babacar Faye, Magatte Ndiaye, Jean Louis Abdourahim Ndiaye.

**Validation:** Isaac Akhenaton Manga, Abdoulkarim Mhadji, Aminata Lam, Fassiatou Tairou, Amadou Seck, Babacar Faye, Magatte Ndiaye, Jean Louis Abdourahim Ndiaye.

**Visualization:** Isaac Akhenaton Manga, Babacar Faye, Magatte Ndiaye, Jean Louis Abdourahim Ndiaye.

**Writing – original draft:** Isaac Akhenaton Manga, Abdoulkarim Mhadji.

**Writing – review & editing:** Aminata Lam, Marie Pierre Diouf, Pab Carole Minlekib, Fassiatou Tairou, Amadou Seck, Babacar Faye, Magatte Ndiaye, Jean Louis Abdourahim Ndiaye.

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
