## [Decision Letter · Decision Letter 0]

17 Oct 2024

PONE-D-24-37026Malaria prevalence in children with history of exposure to seasonal malaria chemoprevention: a cross sectional study in southern SenegalPLOS ONE

Dear Dr. Isaac Manga,

Thank you for submitting your manuscript to PLOS ONE. After careful consideration, we feel that it has merit but does not fully meet PLOS ONE’s publication criteria as it currently stands. Therefore, we invite you to submit a revised version of the manuscript that addresses the points raised during the review process.

**ACADEMIC EDITOR: ****Dear Author**The manuscript titled "Malaria Prevalence in Children with a History of Exposure to Seasonal Malaria Chemoprevention: A Cross-Sectional Study in Southern Senegal" holds significant scientific merit, particularly given the importance of the study's region. However, the reviewers have pointed out several issues that need to be addressed in order to enhance the quality of the manuscript. Thus, several revisions are necessary based on the reviewers' suggestions.

We look forward to receiving your revised manuscript.

Kind regards,

José Luiz Fernandes Vieira

Academic Editor

PLOS ONE

Journal Requirements:

Reviewers' comments:

Reviewer's Responses to Questions

**Comments to the Author**

1. Is the manuscript technically sound, and do the data support the conclusions?

Reviewer #1: Partly

Reviewer #2: Partly

2. Has the statistical analysis been performed appropriately and rigorously? 

Reviewer #1: Yes

Reviewer #2: Yes

3. Have the authors made all data underlying the findings in their manuscript fully available?

Reviewer #1: Yes

Reviewer #2: Yes

4. Is the manuscript presented in an intelligible fashion and written in standard English?

Reviewer #1: Yes

Reviewer #2: No

5. Review Comments to the Author

Reviewer #1: The study titled "Malaria prevalence in children with a history of exposure to seasonal malaria chemoprevention: a cross-sectional study in southern Senegal" addresses a critical issue in malaria control and elimination—the burden of asymptomatic cases or parasite reservoirs that need to be addressed to eliminate the disease. However, significant revisions are required.

Title: The manuscript title should be revised because the definition of a malaria case (fever or history of fever during the last 48 hours with a positive test, such as RDT, blood smear, or PCR) is not clearly stated. Therefore, "malaria prevalence" may not be appropriate for the title.

Methodology:

Study site: The authors provide a brief description of these areas. However, a more detailed account of the malaria epidemiology (incidence or prevalence, transmission season, etc.) is needed.

Sample size calculation: The authors mention, “To detect about 40% of malaria cases, the number of children needed for this study was estimated at 250 using Epi Info.” To determine the sample size, the prevalence or expected prevalence of asymptomatic malaria based on previous studies should be used.

Line 86: How was the history of exposure to SMC evaluated? Did the children have SMC ID cards?

Line 90: The estimated sample size was 250 children. How were these participants distributed across the four regions or health districts (Kédougou, Kolda, Sédhiou, and Tambacounda)? More information is needed on how the participants were selected.

Sociodemographic characteristics of the study population: The distribution of participants by region or health district should be provided.

Line 143: “Prevalence of malaria” should be changed to “Prevalence of asymptomatic malaria.”

Line 148: The authors state that “Among these children (11 years old), 27.6% (95% CI = 17.5-37.7) reported taking SMC in 2013; 19.2% (95% CI = 13.7-24.7) in 2014; and 18.4% (95% CI = 10.3-26.5) in 2015.” Is it realistic to expect 11-year-olds to answer this type of question accurately?

Line 149: “No malaria cases were noted...” should be revised, as this study does not deal with malaria cases.

Additionally, in this section, the distribution of asymptomatic malaria by region or health district should be provided.

Discussion:

Line 164-165: The authors should avoid comparing the prevalence of asymptomatic cases from this study with the national malaria prevalence. The study in Gabon (line 168, ref. 17) targeted malaria cases in adolescents using microscopy diagnosis.

Line 173: The authors state, “Our study also showed that regular use of SMC prevents parasite transmission.” How does it show this?

Conclusion:

Line 196: The authors state, “Extending the SMC target to Senegal could help the country achieve its goal of eliminating malaria by 2030.” However, the results of this study do not fully support such a conclusion.

Moreover, language levels need to be improved

Reviewer #2: Overall comment:There is a need to review the entire manuscript for clarity of the language and for language correction . A review by an English native speaker would help. Additionally, and specially for the discussion and conclusion sections, a carefully review of the statements and related reference checks is needed.

Specific comments:

L31.. The study... not "this"...

L122-123: There were other quantitative variables apart of age? If no please clarify.

L127: Plese specify the age of the children requiring assent...

Results

L13- to 142: A sugestion for improving this descriptin for clarity: "The sex ratio (M/F) of the 226 children included in this study was 1.05. However, gender information was missing for 2.2% of the children. The children were between 11 and 14 years old, with a mean age of 11.9 years (±0.8). The majority were 11 or 12 years old, accounting for 35.4% (95% CI = 29.2-41.6) and 46% (95% CI = 39.5-52.5) of the population, respectively. Age information was missing for 0.4% (95% CI = 0-1.3) of the children.

Most of the children (87.6%; 95% CI = 85.4-89.8) had received SMC in 2014, followed by 38.5% (95% CI = 35.3-41.7) in 2015 and 33.6% (95% CI = 30.5-36.8) in 2013. Children who had received SMC in both 2013 and 2014 accounted for 23% (95% CI = 17.5-28.5), while those who had been exposed in 2013, 2014, and 2015 (3 years) represented 8.4% (95% CI = 4.8-12). Only 0.4% (95% CI = 0-1.3) had received SMC for four consecutive years (2013-2016). (S2 Table).

If the authors agree, the same suggestion may be used for the following sections on results.

L144: Please check the use of singular or plural verbs... It should be "were"...

L145-146: Suggestion to improve for clarity: The number of malaria-positive cases was higher in females, 54.5% (95% CI = 39.8-69.2), compared to males, 43.2% (95% CI = 28.6-57.8).

L150-151: Suggestion for correction in language: " Among the 44 children who tested positive for malaria by PCR..."

L159-163: Apart of improving the text for clarity, it is obvious that if you compare different diagnostic methods the results may be different. I would suggest, if any, relate to studies using the same diagnostic approach or , if not available, explore aditional explanations for the diferences found.

L167-170: From my understanding your study recruited participants with a malaria RDT negative and no symptoms of malaria (which includes fever). Now you are referring to studies with patients positive by microscopy to malaria and with fever. Again, you are comparing diferent methodologies and populations. Please use apropriate references for the discussion.

L175: All referenced studies here (references 3,4,11) are related to effectiveness of the SMC, not "in vivo efficacy"... Please check and re-write accordingly.

L196-199: These a recommendations, not conclusions. Please check and clarify.

6. PLOS authors have the option to publish the peer review history of their article (what does this mean? ). If published, this will include your full peer review and any attached files.

**Do you want your identity to be public for this peer review?** For information about this choice, including consent withdrawal, please see our Privacy Policy .

Reviewer #1: No

Reviewer #2: **Yes: ** Pedro Aide

---

## [Author Response · Author response to Decision Letter 1]

1 Dec 2024

Reviewer #1: The study titled "Malaria prevalence in children with a history of exposure to seasonal malaria chemoprevention: a cross-sectional study in southern Senegal" addresses a critical issue in malaria control and elimination—the burden of asymptomatic cases or parasite reservoirs that need to be addressed to eliminate the disease. However, significant revisions are required.

Title: The manuscript title should be revised because the definition of a malaria case (fever or history of fever during the last 48 hours with a positive test, such as RDT, blood smear, or PCR) is not clearly stated. Therefore, "malaria prevalence" may not be appropriate for the title.

Response: we have changed the title which is now : “Asymptomatic carriage of Plasmodium falciparum in children no longer targeted for chemoprevention of seasonal malaria and with a history of exposure to this strategy: a cross sectional study in southern Senegal.”

Methodology:

Study site: The authors provide a brief description of these areas. However, a more detailed account of the malaria epidemiology (incidence or prevalence, transmission season, etc.) is needed.

Sample size calculation: The authors mention, “To detect about 40% of malaria cases, the number of children needed for this study was estimated at 250 using Epi Info.” To determine the sample size, the prevalence or expected prevalence of asymptomatic malaria based on previous studies should be used.

Response: We rewrote the methodology, explaining the epidemiological aspects that led to the choice of the study area. We also reviewed the sample size calculation, showing that this study was part of a project to evaluate the impact of SMC on the immunity of children who had benefited from this strategy. Therefore, we repeated the calculation of the sample size needed for this study and stated our inclusion criteria.

Line 86: How was the history of exposure to SMC evaluated? Did the children have SMC ID cards?

Response: To assess the history of exposure to SMC, we relied on the parents' memories. Card storage is difficult in rural areas.

Line 90: The estimated sample size was 250 children. How were these participants distributed across the four regions or health districts (Kédougou, Kolda, Sédhiou, and Tambacounda)? More information is needed on how the participants were selected.

Sociodemographic characteristics of the study population: The distribution of participants by region or health district should be provided.

Response: Children were recruited from different regions on the basis of the case-control study.

Line 143: “Prevalence of malaria” should be changed to “Prevalence of asymptomatic malaria.”

Response: We changed it by Asymptomatic carriage of Plasmodium falciparum

Line 148: The authors state that “Among these children (11 years old), 27.6% (95% CI = 17.5-37.7) reported taking SMC in 2013; 19.2% (95% CI = 13.7-24.7) in 2014; and 18.4% (95% CI = 10.3-26.5) in 2015.” Is it realistic to expect 11-year-olds to answer this type of question accurately?

Response: We modified the sentence to show that the parents who answered the questions

Line 149: “No malaria cases were noted...” should be revised, as this study does not deal with malaria cases.

Additionally, in this section, the distribution of asymptomatic malaria by region or health district should be provided.

Response: No parasite carriage was noted among those exposed in 2013, 2014, 2015 and 2016 is the sentence we proposed and we also gave the distribution of Plasmodium carriage by regions as you asked.

Discussion:

Line 164-165: The authors should avoid comparing the prevalence of asymptomatic cases from this study with the national malaria prevalence. The study in Gabon (line 168, ref. 17) targeted malaria cases in adolescents using microscopy diagnosis.

Line 173: The authors state, “Our study also showed that regular use of SMC prevents parasite transmission.” How does it show this?

Response: We have rewritten the discussion to reflect your recommendations, which we believe are relevant. You can see this in the newly submitted version.

Conclusion:

Line 196: The authors state, “Extending the SMC target to Senegal could help the country achieve its goal of eliminating malaria by 2030.” However, the results of this study do not fully support such a conclusion.

Response: Here is the new conclusion of the study: This study found significant asymptomatic Plasmodium falciparum carriage in adolescents with a history of exposure to SMC. It was more pronounced in children who had moved from the target group of this strategy in Senegal. Older children are therefore a major reservoir of parasites, maintaining malaria transmission in the country.

Moreover, language levels need to be improved:

Response : As you will see in the new version, we asked a native english speaker to help us improve the language.

Reviewer #2: Overall comment:There is a need to review the entire manuscript for clarity of the language and for language correction . A review by an English native speaker would help. Additionally, and specially for the discussion and conclusion sections, a carefully review of the statements and related reference checks is needed.

Response: We asked a native English speaker to help us improve the language.

Specific comments:

L31. The study... not "this"...

Response: We have rewritten the sentence to reflect your recommendation.

L122-123: There were other quantitative variables apart of age? If no please clarify.

Response: We reworded the sentence in the data analysis section to make it clearer, as the children's age was the only quantitative variable.

L127: Please specify the age of the children requiring assent...

Response: The assent of all the children (11-14 years old) included had also been obtained before the start of the study.

Results

L13- to 142: A sugestion for improving this descriptin for clarity: "The sex ratio (M/F) of the 226 children included in this study was 1.05. However, gender information was missing for 2.2% of the children. The children were between 11 and 14 years old, with a mean age of 11.9 years (±0.8). The majority were 11 or 12 years old, accounting for 35.4% (95% CI = 29.2-41.6) and 46% (95% CI = 39.5-52.5) of the population, respectively. Age information was missing for 0.4% (95% CI = 0-1.3) of the children.

Most of the children (87.6%; 95% CI = 85.4-89.8) had received SMC in 2014, followed by 38.5% (95% CI = 35.3-41.7) in 2015 and 33.6% (95% CI = 30.5-36.8) in 2013. Children who had received SMC in both 2013 and 2014 accounted for 23% (95% CI = 17.5-28.5), while those who had been exposed in 2013, 2014, and 2015 (3 years) represented 8.4% (95% CI = 4.8-12). Only 0.4% (95% CI = 0-1.3) had received SMC for four consecutive years (2013-2016). (S2 Table).

If the authors agree, the same suggestion may be used for the following sections on results.

L144: Please check the use of singular or plural verbs... It should be "were"...

L145-146: Suggestion to improve for clarity: The number of malaria-positive cases was higher in females, 54.5% (95% CI = 39.8-69.2), compared to males, 43.2% (95% CI = 28.6-57.8).

L150-151: Suggestion for correction in language: " Among the 44 children who tested positive for malaria by PCR..."

Response: To take account of your suggestions on how to present the results, the entire results section has been rewritten.

Discussion section

L159-163: Apart of improving the text for clarity, it is obvious that if you compare different diagnostic methods the results may be different. I would suggest, if any, relate to studies using the same diagnostic approach or , if not available, explore aditional explanations for the diferences found.

L167-170: From my understanding your study recruited participants with a malaria RDT negative and no symptoms of malaria (which includes fever). Now you are referring to studies with patients positive by microscopy to malaria and with fever. Again, you are comparing diferent methodologies and populations. Please use apropriate references for the discussion.

L175: All referenced studies here (references 3,4,11) are related to effectiveness of the SMC, not "in vivo efficacy"... Please check and re-write accordingly.

Response: We have modified our discussion to take account of your recommendations, i.e. comparison with the same population and the same techniques used to test for Plasmodium. Our references have therefore been modified

L196-199: These recommendations, not conclusions. Please check and clarify.

Response: Here is the new conclusion of the study: This study found significant asymptomatic Plasmodium falciparum carriage in adolescents with a history of exposure to SMC. It was more pronounced in children who had moved from the target group of this strategy in Senegal. Older children are therefore a major reservoir of parasites, maintaining malaria transmission in the country.

---

## [Decision Letter · Decision Letter 1]

9 Jan 2025

Asymptomatic carriage of Plasmodium falciparum in children no longer targeted for chemoprevention of seasonal malaria and with a history of exposure to this strategy: a cross sectional study in southern Senegal.

PONE-D-24-37026R1

Dear Dr. Isaac Manga

We’re pleased to inform you that your manuscript has been judged scientifically suitable for publication and will be formally accepted for publication once it meets all outstanding technical requirements.

Kind regards,

José Luiz Fernandes Vieira

Academic Editor

PLOS ONE

Additional Editor Comments 

After carefully reviewing the changes made to the manuscript, I recommend its publication in *PLOS ONE* congratulations

José Luiz Vieira

Reviewers' comments:

**Comments to the Author**

1. If the authors have adequately addressed your comments raised in a previous round of review and you feel that this manuscript is now acceptable for publication, you may indicate that here to bypass the “Comments to the Author” section, enter your conflict of interest statement in the “Confidential to Editor” section, and submit your "Accept" recommendation.

Reviewer #1: All comments have been addressed

2. Is the manuscript technically sound, and do the data support the conclusions?

Reviewer #1: Yes

3. Has the statistical analysis been performed appropriately and rigorously? 

Reviewer #1: Yes

4. Have the authors made all data underlying the findings in their manuscript fully available?

Reviewer #1: Yes

5. Is the manuscript presented in an intelligible fashion and written in standard English?

Reviewer #1: Yes

6. Review Comments to the Author

Reviewer #1: (No Response)

7. PLOS authors have the option to publish the peer review history of their article (what does this mean? ). If published, this will include your full peer review and any attached files.

**Do you want your identity to be public for this peer review?** For information about this choice, including consent withdrawal, please see our Privacy Policy .

Reviewer #1: No

---

## [Editor Report · Acceptance letter]

PONE-D-24-37026R1

PLOS ONE

Dear Dr. Manga,

I'm pleased to inform you that your manuscript has been deemed suitable for publication in PLOS ONE. Congratulations! Your manuscript is now being handed over to our production team.

Kind regards,

on behalf of

Dr. José Luiz Fernandes Vieira

Academic Editor

PLOS ONE